# Direct imaging of structural changes induced by ionic liquid gating leading to engineered three-dimensional meso-structures

Bin Cui[1], Peter Werner[1], Tianping Ma[1], Xiaoyan Zhong[2], Zechao Wang [2,3], James Mark Taylor[1], Yuechen Zhuang[1] & Stuart S.P. Parkin [1]

The controlled transformation of materials, both their structure and their physical properties, is key to many devices. Ionic liquid gating can induce the transformation of thin-film materials over long distances from the gated surface. Thus, the mechanism underlying this process is of considerable interest. Here we directly image, using in situ, real-time, high-resolution transmission electron microscopy, the reversible transformation between the oxygen vacancy ordered phase brownmillerite $SrCoO_{2.5}$ and the oxygen ordered phase perovskite $SrCoO_3$. We show that the phase transformation boundary moves at a velocity that is highly anisotropic, traveling at speeds ~30 times faster laterally than through the thickness of the film. Taking advantage of this anisotropy, we show that three-dimensional metallic structures such as cylinders and rings can be realized. Our results provide a roadmap to the construction of complex meso-structures from their exterior surfaces.

[1] Max Planck Institute for Microstructure Physics, Halle 06120, Germany. [2] Beijing National Center for Electron Microscopy, Laboratory of Advanced Materials and Department of Materials Science and Engineering, Tsinghua University, Beijing 100084, China. [3] Ernst Ruska-Centre for Microscopy and Spectroscopy with Electrons Research Centre Jülich, Jülich D-52425, Germany. Correspondence and requests for materials should be addressed to S.S.P.P. (email: stuart.parkin@mpi-halle.mpg.de)

The use of electrical methods to manipulate the ground states of correlated-electron oxides has much potential for the development of electronic and ionitronic devices[1–11]. One method is to take advantage of the very high electric fields that are generated by polarizing ionic liquids (IL). This method has enabled, for example, the reversible transformation of thin oxide films between insulating and metallic[1,4,8–10,12] or super-conducting states[6]. In some cases it has been shown that these IL gate-induced transformations are predominantly the result of electric-field-induced structural changes that rely on the migration of oxygen[4,10]. However, the detailed mechanisms underlying these processes are poorly understood. Thus, the visualization of oxygen migration under IL gating by direct observation at the atomic level is of great interest. In situ high-resolution transmission electron microscopy (TEM) is an excellent technique for such studies, but a major challenge is that the IL strongly degrades the image quality due to scattering of the electron beam[13]. We overcome this obstacle by positioning a droplet of the IL close to, but not within, the imaged region of the TEM lamellae by the use of an atomic force microscope.

For these studies, we chose to explore two phases within the strontium cobaltite family ($SrCoO_x$, SCO), namely the perovskite (P) phase $SrCoO_3$ and the brownmillerite (BM) phase $SrCoO_{2.5}$. These phases have been extensively studied: $SrCoO_{2.5}$ is an antiferromagnetic insulator, whereas $SrCoO_3$ is a ferromagnetic metal[14]. Here, we find direct evidence for massive oxygen migration induced by liquid gating of $SrCoO_x$ thin films. We find that the migration takes place over several minutes and that its time-scale is an order of magnitude faster parallel, as compared to perpendicular to the film surface. Taking advantage of these findings we create a series of three-dimensional meso-structures in $SrCoO_x$ and several other oxide films.

## Results

**Experimental setup for in situ TEM IL gating.** In these studies, $SrCoO_{2.5}$ films, 40 nm thick, were prepared by pulsed laser deposition (PLD) on (001) $SrTiO_3$ (STO) and 0.5 wt% Nb-doped $SrTiO_3$ (NSTO) substrates. A single STO wafer was used to prepare samples for: in situ TEM and STEM (scanning transmission electron microscopy), ex situ STEM (samples ST-A and -B), and transport measurements, while samples on NSTO were used for meso-structure fabrication and conductive atomic force microscope (CAFM) measurements. Lamellae for the TEM studies were prepared using standard thinning techniques and supported on an in situ chip holder with electrodes for the application of a gate voltage ($V_G$). As schematically shown in Fig. 1, a droplet of the IL was positioned to cover both the sample and a lateral gate electrode that formed part of a field effect transistor (FET) (see Fig. 1a and more details in Methods and Supplementary Fig. 1 and 2). The channel of the FET is formed from a thin film of $SrCoO_{2.5}$. While applying a voltage to the gate electrode, reversible changes between the BM $SrCoO_{2.5}$ and the perovskite $SrCoO_3$ phases could be contemporaneously observed in the lamellae (Fig. 1b). Indeed, consistent with our TEM observations, the film is converted from an initially insulating state to a final metallic state (see Fig. 1c).

**In situ observation of structural changes induced by gating.** The structure of the BM $SrCoO_{2.5}$ is an oxygen vacancy ordered phase that is derived from the perovskite phase $SrCoO_3$ by removal of one oxygen atom in every other $CoO_2$ sublayer. The crystal lattice of the pristine $SrCoO_{2.5}$ sample, as shown in the high-resolution TEM image in Fig. 2a, is characterized by a modulated structure (unit-cell doubling along the $c$-direction). This modulation is also seen in corresponding superlattice reflections in the FFT (fast Fourier transform) diffractograms of the TEM lattice plane images, as indicated by small yellow circles in Fig. 2b. After the application of $V_G = -3$ V for 5 min, the modulated structure in the area near the surface of the thin film changes to the perovskite structure, consistent with the weaker superlattice reflections in the FFT diffractograms (Fig. 2c, d). This is consistent with injection of oxygen into the material, resulting

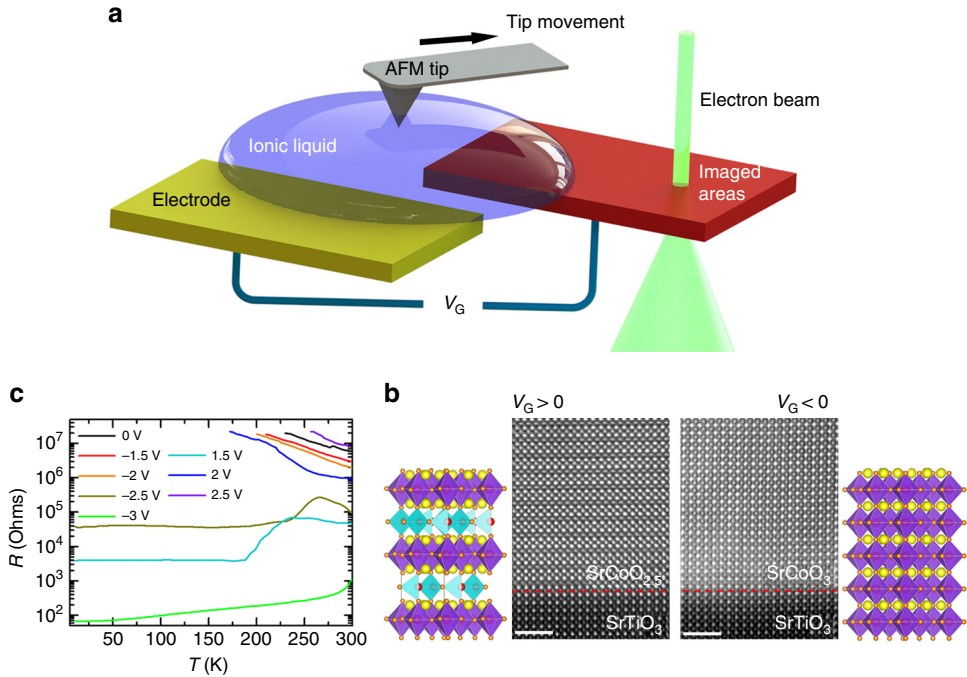

**Fig. 1** Ionic liquid gating effect for $SrCoO_x$. **a** Schematic drawing of in situ TEM measurement setup with ionic liquid placement by an atomic force microscope tip. Positive and negative gate voltages $V_G$ are applied to the ionic liquid by the gate electrode. **b** Typical STEM-HAADF images are shown for the as-deposited film $SrCoO_{2.5}$ and the same film after gating with $V_G = -3$ V for 15 min. The white scale bar corresponds to 2 nm. The yellow, red, and orange spheres denote Sr, Co, and O, respectively. **c** Resistance versus temperature curves at various $V_G$ as indicated in the figure for a 40 nm thick $SrCoO_{2.5}$ film

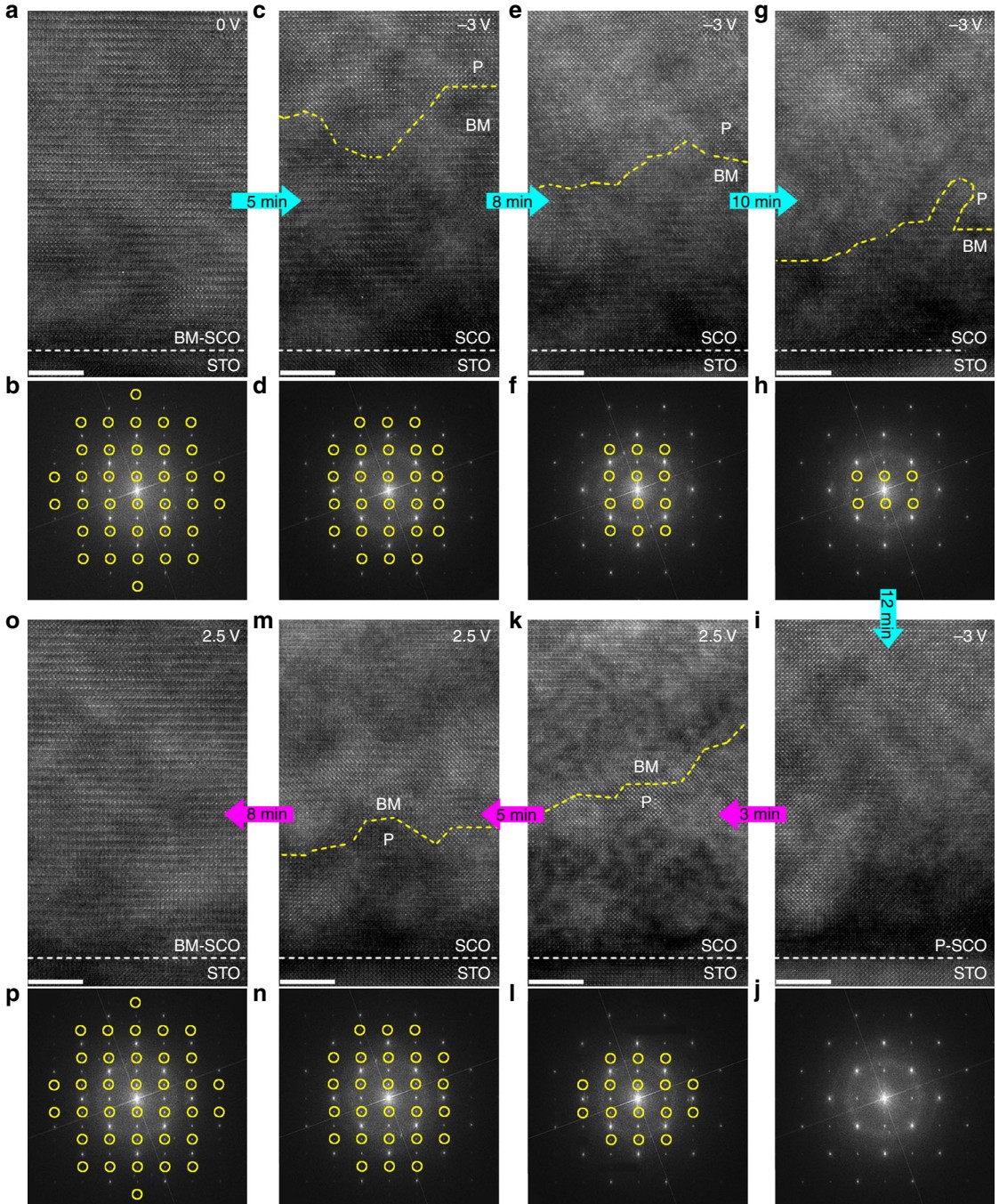

**Fig. 2** In situ TEM results of $SrCoO_x$ under ionic liquid gating. TEM images and corresponding FFT diffractograms for $SrCoO_x$ (SCO)/(001) $SrTiO_3$ (STO) after the application of various gate voltages and gating times: **a**, **b** $V_G = 0\,V$, 0 min; **c**, **d** $V_G = -3\,V$, 5 min; **e**, **f** $V_G = -3\,V$, 8 min; **g**, **h** $V_G = -3\,V$, 10 min; **i**, **j** $V_G = -3\,V$, 12 min; **k**, **l** $V_G = +2.5\,V$, 3 min; **m**, **n** $V_G = +2.5\,V$, 5 min; **o**, **p** $V_G = +2.5\,V$, 8 min. The white scale bars in TEM images are 5 nm. The yellow and white dashed lines in the TEM images indicate the boundary between brownmillerite (BM) and the perovskite (P) SCO phases, and the interface between SCO and STO, respectively. The yellow circles in the FFTs denote the superlattice reflections resulting from the modulated structure in the brownmillerite phase. The cyan and magenta arrows indicate the sequence of images taken. Cyan corresponds to $V_G = -3\,V$ and magenta to $V_G = +2.5\,V$

in the transition: $SrCoO_{2.5} \rightarrow SrCoO_3$. The yellow dashed lines in the TEM images in Fig. 2 mark the approximate boundary between the BM and the perovskite SCO phases. With increased gating time (8 and 10 min in Fig. 2e–h, respectively), the $SrCoO_{2.5} \rightarrow SrCoO_3$ transition evolves from the surface to the bottom of the thin film. This phase transition is completed after ~12 min (Fig. 2i, j). Energy dispersive x-ray spectroscopy confirms a significant increase in oxygen content as a result of the

gating process that is also clearly non-volatile (see Supplementary Figs. 3 and 4).

Subsequently, we applied a reversed polarity gate voltage $V_G = +2.5\,V$. As shown in Fig. 2k–p, when the gating time is increased, the modulated structure first appears at the surface and then gradually extends to the bottom of the film, accompanied by enhanced superlattice reflections. This is direct evidence that $V_G = +2.5\,V$ creates oxygen vacancies ($V_O$) in the sample,

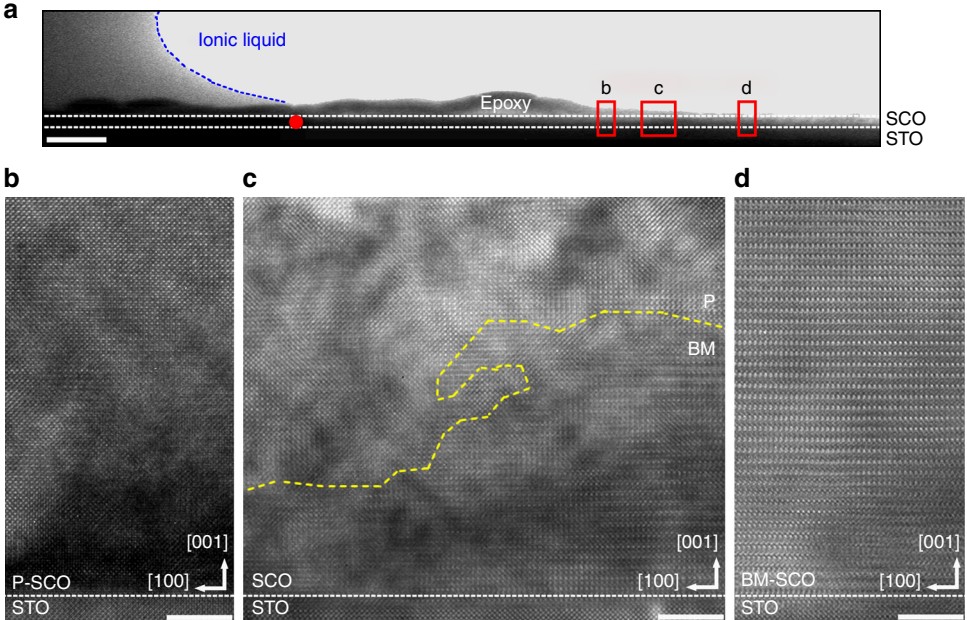

**Fig. 3** Distance-dependent ionic liquid gating effect. **a** Overview of in situ TEM sample, where the selected areas for high magnification images are marked by red boxes. The while scale bar in **a** corresponds to 200 nm. The edge of the ionic liquid (IL) is indicated by a blue dashed line. The distances between the ionic liquid and selected areas are estimated with respect to the red point and the centers of the red frames. The TEM images of selected areas at different distances from the edge of the IL, after gating at $V_G = -3$ V for 12 min: **b** $d \approx 1.03$ μm; **c** $d \approx 1.19$ μm; and **d** $d \approx 1.51$ μm. The interface between the SCO film and the STO substrate is shown as a white dashed line. The yellow dashed line in **c** shows the boundary between brownmillerite (BM) and perovskite (P) SCO phases. The white scale bars in **b**–**d** are 5 nm

resulting in the phase transition SrCoO$_3$ → SrCoO$_{2.5}$. Such a reversible, non-volatile, electrical manipulation between the two SCO phases, unambiguously demonstrates the role of $V_O$ in the IL gating process. This phase transition is also confirmed by X-ray diffraction (XRD) from large area thin-film samples and $V_G$ dependent resistance curves measured in FET devices (see Supplementary Figs. 5 and 6)[15]. We note that the IL gate-controlled phase transition is reproducible, and any electron beam irradiation effects can be ruled out (see Supplementary Figs . 7 and 8). The reversible phase transition that extends from the surface to the bottom of the film can be followed more closely by depth-dependent line-scan FFT diffractograms (see Supplementary Fig. 9). From Fig. 2c–i, the phase boundary moves ~20.3 nm over a 7 min period when $V_G = -3$ V, while from Fig. 2k–o, the phase boundary moves ~18.0 nm over 5 min when $V_G = +2.5$ V. The speed of the vertical phase transition is thus estimated to be ~2.9 nm min$^{-1}$ for SrCoO$_{2.5}$ → SrCoO$_3$, and ~3.6 nm min$^{-1}$ for SrCoO$_3$ → SrCoO$_{2.5}$.

In addition to the vertical oxygen transport across the thin film, lateral oxygen transport within the thin film also takes place. This can be seen by inspecting TEM images of the film at different lateral distances $d$ from the IL droplet, as shown in Fig. 3a. Images taken at $d \approx 1.03$, 1.19, and 1.51 μm are shown in Fig. 3b–d, respectively, after $V_G = -3$ V was applied for 12 min. The area in Fig. 3b is very close to that shown in Fig. 2, where the sample has been fully converted to SrCoO$_3$. As $d$ is increased to ~1.19 μm, both SCO phases are found and the phase boundary is marked by a yellow dashed line in Fig. 3c. Remarkably, the SrCoO$_3$ phase occupies a much larger area closer to the IL and near the film surface, indicating that oxygen ions migrate first from the IL towards the other end of the thin film through the surface. In contrast, at $d \approx 1.51$ μm (Fig. 3d), the structure appears to be unaffected by $V_G$. As distinct from the vertical oxygen transport, the lateral oxygen transport displays a much higher velocity of ~10$^2$ nm min$^{-1}$. This high speed oxygen transport could be attributed to a surface "fast oxygen" transport lane, where the

decreased number of bonds and reconstruction phenomena will considerably reduce the activation energy for atom jump processes[16]. Note that the lateral migration rate of O$^{2-}$ that we observe at room temperature in SCO is comparable to that of some cathode materials in solid oxide fuel cell (SOFC) but which occurs at very high temperatures of ~700 °C[17,18].

**Oxygen vacancy distribution and electronic structure**. STEM-HAADF (high angle annular dark field) and STEM-ABF (annular bright-field) measurements were used to further probe the distribution of oxygen ions or vacancies. Due to the non-volatility of the IL gating effects[4,8–10], ex situ gated samples were used to obtain higher resolution images. Two cross-sectioned samples were prepared for these studies: sample ST-A was first gated at $V_G = -3$ V followed by a second gating at $V_G = +2.5$ V, while sample ST-B was gated at $V_G = -3$ V. The gating time for each $V_G$ was 30 min. For these ex situ experiments, the samples were [110] oriented in-plane so as to better observe the ordering of the $V_O$ channels along [1–10] (for the in situ experiments discussed above the sample was [100] oriented in-plane). The HAADF images show that the structure of ST-A consists of alternate stacking of fully oxygenated octahedral (SrO–CoO$_2$–SrO) and oxygen-deficient tetrahedral (SrO–CoO–SrO) sublayers, whereas ST-B has a uniform perovskite structure (compare Fig. 4a, b). Another clear distinction between the two samples is the pairing of the Co atoms within the tetrahedral sublayers in sample ST-A resulting from the ordering of the oxygen vacancies that can be clearly seen in Fig. 4a.

The ABF images taken concurrently with the HAADF images, exhibit an opposite contrast and are more sensitive to light elements[19]. In Fig. 4c, d, Sr, Co, O, and $V_O$ sites in the ABF images are indicated by yellow, red, and orange spheres, and orange circles, respectively. By comparing the ABF images for samples ST-A and -B, one can clearly see the columns of $V_O$, which are located between the Co–Co pairs (orange circles in

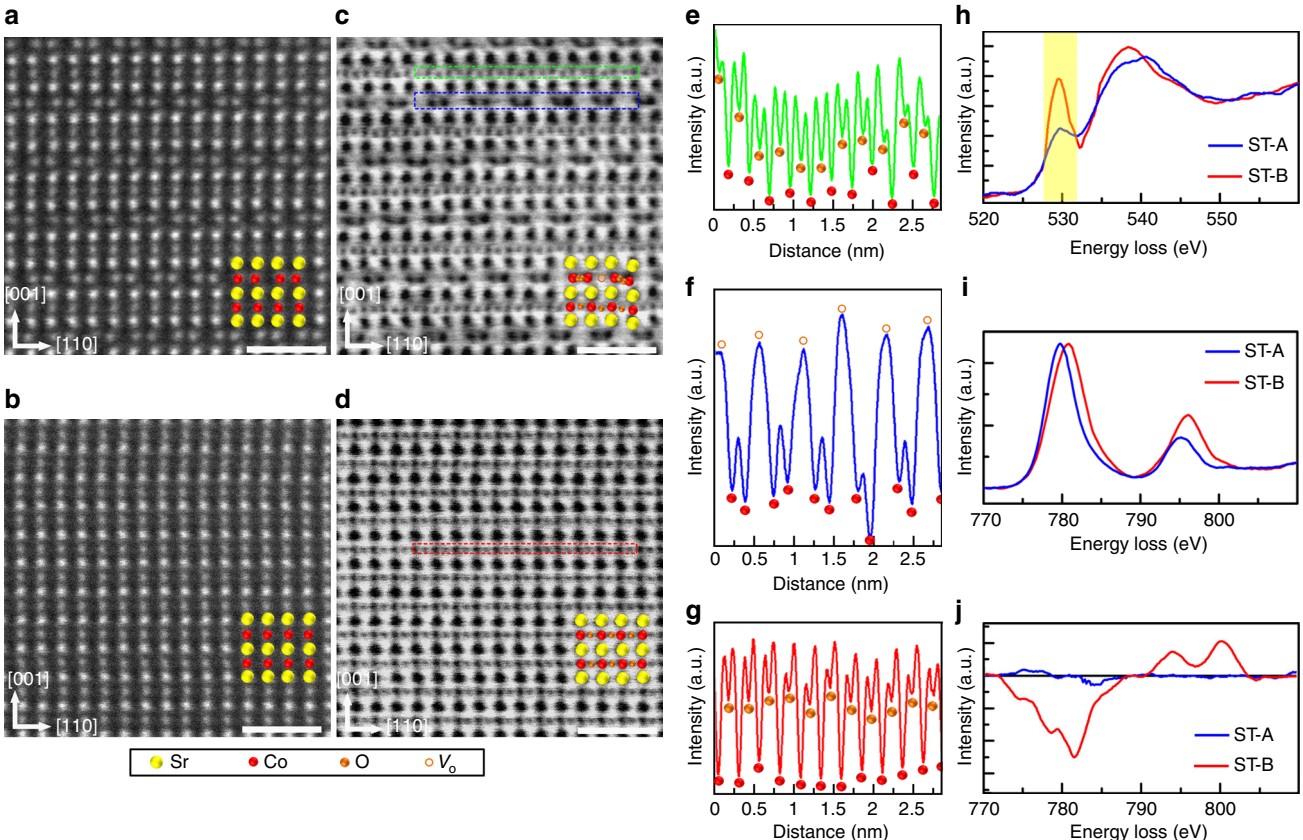

**Fig. 4** Crystal and electronic structure of ex situ gated samples. STEM-HAADF images of **a** Sample ST-A and **b** Sample ST-B. STEM-ABF images of **c** Sample ST-A and **d** Sample ST-B, respectively. The higher the atomic number, the brighter (darker) the atom in HAADF (ABF). The white scale bars in **a–d** are 1 nm. Sr, Co, O, and $V_O$ sites are indicated by yellow, red, and orange spheres, and orange circles, respectively. Intensity line-scans of the ABF images of **e** $CoO_2$ and **f** CoO sublayers in Sample ST-A, and **g** $CoO_2$ sublayer in Sample ST-B. The line-scans correspond to the green, blue and red boxes as indicated in the ABF images, respectively. **h** O-K EELS; **i** Co-L EELS; and **j** Co-EMCD of Sample ST-A and -B. The pre-peak in the O K-edge is highlighted in yellow

Fig. 4c). This is also reflected in the line-scans along the $CoO_2$ and CoO planes shown for ST-A (Fig. 4e, f) and the $CoO_2$ planes for sample ST-B (Fig. 4g), where the Co and O sites are valleys (Fig. 4e, g) while $V_O$ sites are the higher peaks (Fig. 4f).

Beyond changes in crystal structure, the electronic structure and magnetic properties are also dramatically affected by IL gating. In electron energy-loss spectroscopy (EELS) at the O K-edge (Fig. 4h), the pre-peak intensity of sample ST-B is higher than that of ST-A due to the stronger Co 3d–O 2p hybridization at the higher Co oxidation state[20]. Meanwhile, compared with ST-A, the Co L-edge of ST-B in Fig. 4i shifts to a ~1.0 eV higher energy with a smaller $L_3/L_2$ intensity ratio ($L_3/L_2 = 2.84$ for ST-A, and 1.96 for ST-B), consistent with that for expected for $Co^{3+}$ and $Co^{4+}$ (ref. [21]). Such a change in electronic structure is confirmed by X-ray absorption spectroscopy (XAS) (see Supplementary Fig. 10). These crystal and electronic structure changes show convincingly that ~0.5 oxygen per formula unit (~17%) is added or removed during the IL gating process in ST-A ($SrCoO_{2.5}$) and ST-B ($SrCoO_3$), respectively. Note that such a massive oxygen addition and removal is much higher than those found in earlier studies of the IL gating of other oxides: ~3% in $VO_2$[4] and $WO_3$[9], ~5% in $SmNiO_3$[5] and $(La,Sr)MnO_3$[8]. Moreover, an electron energy-loss magnetic chiral dichroism (EMCD) signal at the Co L-edge is clearly seen in sample ST-B (see Fig. 4g and Supplementary Fig. 11) but not in sample ST-A[22,23], clearly indicating the ferromagnetic character of ST-B. The magnetic property changes caused by IL gating are also confirmed by magnetization studies on thin-film samples (see Supplementary

Fig. 12), which are consistent with ferromagnetic and antiferromagnetic ground states in $SrCoO_{2.5}$ and $SrCoO_3$, respectively, as found in the bulk[14].

**IL gate engineered meso-structures.** The multi-directional oxygen transport, that was directly observed above, allows for the possibility of creating complex meso-structures by local IL gating. To demonstrate this possibility, we have created patterns of orifices in resist layers that were spin-coated onto several distinct oxide films that show different degrees of anisotropic oxygen ion transport. Here we present data on IL gated (001) oriented $VO_2$, $La_{0.45}Sr_{0.55}MnO_3$, and $SrCoO_{2.5}$ films that were each covered by resist layers, patterned using electron beam lithography to have orifices 1 μm in diameter, spaced 2 μm apart (see Supplementary Figs. 13 and 14). The resist is covered by the IL but the oxide films only "see" IL within the orifices. Thus, on gating, an electric double layer can only be formed within these limited regions at the oxide surface. All the samples could be changed from a high-resistance state to a low-resistance state by gating (see Supplementary Fig. 15)[4,10,24]. After gating, the IL and resist are removed and CAFM imaging is carried out to profile the conductivity of the films within and without the orifices.

Typical CAFM results are shown in Fig. 5a–c. The films of (001) $VO_2$ display a metal–insulator transition above room temperature (see Supplementary Fig. 15) so the films are poorly conducting at ambient temperature. After IL gating we find clear evidence that only within the orifice is there a change in the film

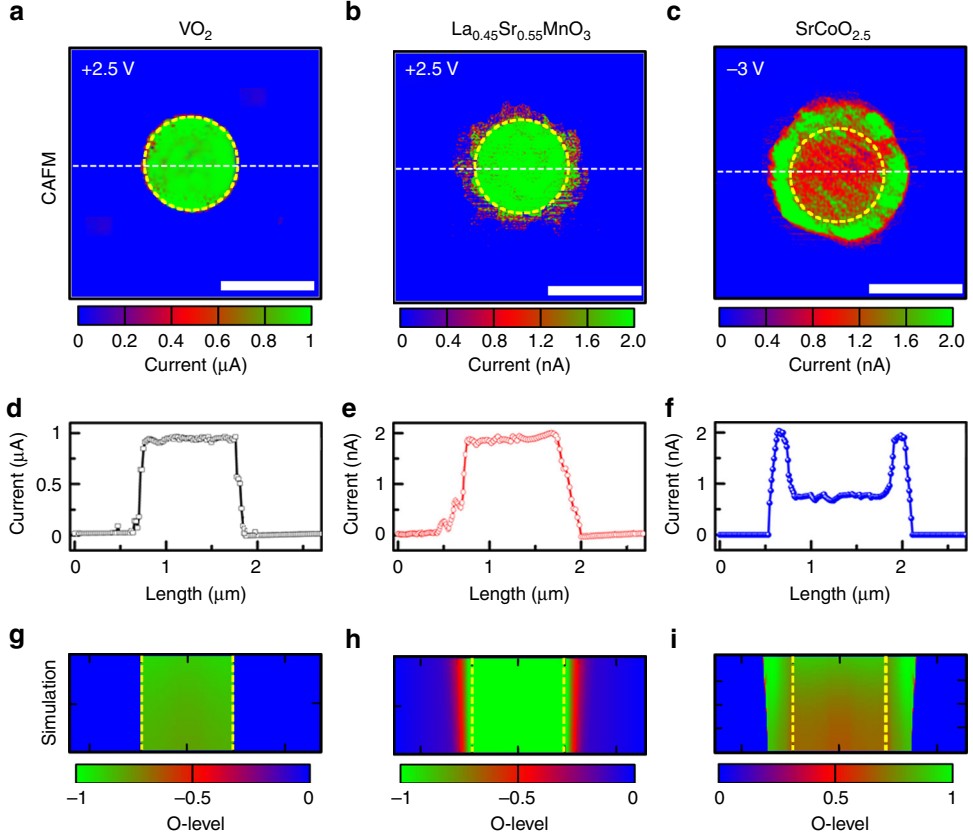

**Fig. 5** 3D meso-structures formed by ionic liquid gating. Typical CAFM images for **a** $VO_2$, **b** $La_{0.45}Sr_{0.55}MnO_3$, and **c** $SrCoO_{2.5}$ after ionic liquid gating through orifices in the resist overlaid on the sample. Gate voltages of $+2.5$ V for $VO_2$ and $La_{0.45}Sr_{0.55}MnO_3$ and $-3$ V for $SrCoO_{2.5}$ films were applied for 15 min. The IL and resist were then removed before CAFM measurements were carried out. Typical CAFM images of a single orifice are shown. The scale bars in **a**–**c** are 1 μm. **d**–**f** are 1D current profiles taken at positions indicated by the white dashed lines in **a**–**c**, respectively. Simulations of the oxygen levels (O-levels) for **g** $VO_2$, **h** $La_{0.45}Sr_{0.55}MnO_3$, and **i** $SrCoO_{2.5}$ after ionic liquid gating are shown for the same size region as the CAFM images. The yellow dashed lines indicate where either $O^{2-}$ ($V_G = -3$ V) or $V_O$ ($V_G = +2.5$ V) enters the sample. The O-level is indicated by the color gradient, as shown below each simulation. Note that color gradient is reversed for SCO as compared to $VO_2$ and $La_{0.45}Sr_{0.55}MnO_3$

conductivity, with a substantial increase in conductivity: the regions covered by resist are not affected at all by the IL gating (Fig. 5a). These results are consistent with the highly anisotropic oxygen transport in $VO_2$ that only takes place along [001] oriented channels[25], so that, for these (001) oriented films, there is little lateral transport. In contrast, oxygen transport in $La_{0.45}Sr_{0.55}MnO_3$ is known to be take place along all three principal crystallographic directions[17,18]. In this case we find that the initially highly resistant film becomes conducting not only within, but well beyond the circumference of the orifices (Fig. 5b). Finally, for the films of $SrCoO_{2.5}$ a more complex conductivity pattern is found after gating with increases in conductivity well beyond the orifice as for $La_{0.45}Sr_{0.55}MnO_3$, but with a ring of highest conductivity around the edge of the orifice (Fig. 5c). The distinct conductivity patterns formed within these three oxide materials are readily compared in the 1D conductivity profiles shown in Fig. 5d–f. The evolution of the conductivity patterns with gating time are shown in Supplementary Fig. 16.

We propose that the ring of highest conductivity in $SrCoO_{2.5}$ is caused by the accumulation of oxygen at the interface between the original BM phase and the IL gate-induced perovskite phase, which is consistent with high-resolution cross-section STEM-HAADF imaging (see Supplementary Fig. 17). Two-dimensional finite element diffusion simulations were carried out to explore the origin of the IL gate-induced ring structure. The simulations are carried out within the film's cross-section. We suppose that IL gating results in gradients in oxygen level (O-level), both

perpendicular and parallel to the film surface, that will then lead to, via Fick's second law, diffusion of oxygen. In the simulations, for $VO_2$ and $La_{0.45}Sr_{0.55}MnO_3$, $V_O$ are injected, whereas for SCO, $O^{2-}$ are injected through the orifices at a constant rate. In addition, we include a phase boundary between the BM and perovskite phases and we assume that oxygen can diffuse across this boundary only when the O-level difference across this interface exceeds a threshold value. Moreover, at the phase boundary there is likely band bending, which would account for a local excess or deficiency in oxygen concentration[26], leading to a modification of diffusion coefficients at the phase boundary (see Supplementary Fig. 18 and Supplementary Table 1 for more details). Typical results of this model in Fig. 5g–i are compared with the CAFM conductivity maps in Fig. 5a–c. As can be seen in Fig. 5g–i the simulations can qualitatively describe the distinct conductivity patterns for the three different oxide films. The influence of simulation parameters on the O-level is discussed in Supplementary Fig. 19. By decreasing the pattern dimensions to the nanoscale, such a method of patterned IL gating has the potential for the construction of a wide range of mesoscopic devices, such as tunneling junctions, and quantum wells, wires, and dots.

## Discussion

We have shown that in situ TEM studies of samples subject to ion liquid gating is made possible by the manipulation of droplets of

the IL using a standard AFM on a suitably prepared device. The very low vapor pressure makes it unnecessary to contain the IL in the TEM column unlike most other liquids. Applying this technique we observe the real-time transformation of the ordered oxygen vacancy structure of $SrCoO_{2.5}$ to its fully oxygenated form $SrCoO_3$. This requires a massive influx of oxygen, even though the in situ gating experiments are carried out in the high vacuum environment of the TEM column (which has a pressure of $\sim 4 \times 10^{-9}$ mbar). This solves a longstanding controversy about the origin of the oxygen which must therefore be dissolved in the IL itself. One of the most surprising results of our studies is the highly anisotropic velocity of the phase front between the initial and final phases induced by IL gating. In both gating directions the phase front moves $\sim 30$ times faster along the oxide thin-film surface than perpendicular to the surface, which is attributed to the distinct properties of the surface.

By taking advantage of the IL gate-induced anisotropic oxygen motion we show that complex three-dimensional meso-structures, such as cylinders and rings, can be created from an exterior surface. Using simulations we show that these structures result from a combination of anisotropic oxygen migration and critical concentration oxygen gradients below which migration is suppressed. Thus, our results open a path to the formation and dynamic manipulation of complex three-dimensional structures within an interior volume that are formed by IL gating at the exterior surfaces. Such an approach could be used to develop a number of ionitronic devices, for example, for neuromorphic computing applications, especially reservoir computing approaches.

## Methods

**Sample preparation.** $SrCoO_{2.5}$ thin films, 40 nm thick, were grown on $10 \times 10$ mm$^2$ STO and NSTO (001) substrates at 750 °C, in an oxygen pressure ($p_O$) of $5 \times 10^{-4}$ mbar, using PLD. Twenty nanometer thick (001) oriented $VO_2$ films were grown at 400 °C on 0.5 wt% Nb-doped $TiO_2$ substrate at $p_O = 1.9 \times 10^{-2}$ mbar, while 20 nm thick (001) oriented $La_{0.45}Sr_{0.55}MnO_3$ films were grown at 600 °C on NSTO with $p_O = 2.0 \times 10^{-1}$ mbar. After deposition, the $SrCoO_{2.5}$ and $VO_2$ films were cooled to room temperature in the same oxygen pressure as use for the growth, whereas 500 mbar $O_2$ was used for cooling the $La_{0.45}Sr_{0.55}MnO_3$ films. All the data presented in this article are from samples prepared from the same wafer in each case, except for the CAFM results. The IL $N,N$-diethyl-$N$-(2-methoxyethyl)-$N$-methylammonium bis(trifluoromethylsulfonyl)-imide (DEME-TFSI), was used for all gating experiments. The IL and the devices were separately baked at 130 °C in high vacuum ($10^{-7}$ mbar) for at least 12 h before the gating experiments were carried out.

Transistor devices for transport measurements were prepared by photolithography and wet etching in the form of Hall-bars with lateral gate electrodes located in the vicinity of the channel[8]. The channel is 400 µm long and 100 µm wide. Electrical contacts to the edge of the channel were formed from Au (60 nm)/Cr (10 nm) that was deposited by thermal evaporation. For the ex situ STEM, SQUID, XRD, and XAS studies, large area thin films, $\sim 5 \times 2.5$ mm$^2$ in size, were prepared by gating in a probe station (pressure $< 10^{-6}$ mbar) for 30 min[4,8,9]. The ST-A samples were gated in the sequence, $V_G = -3$ V, then $V_G = +2.5$ V, while the samples ST-B were gated using $V_G = -3$ V. Each $V_G$ was applied for 30 min. The preparation of the transistor devices and those for ex situ measurements are shown schematically in Supplementary Fig. 20.

The resist pattern arrays of orifices 1 µm in diameter, spaced 2 µm apart (positive resist ZEP520A $\sim 400$ nm thick), were exposed by electron beam lithography (Raith Nanofabrication system). In the gating process only the areas in the orifices are in direct contact with the IL. The edge of the samples was surrounded by paraffin, hence the leakage current from IL to the conductive substrate was avoided in the gating process. For the orifice experiments, $V_G = -3$ V was applied to $VO_2$ and $La_{0.45}Sr_{0.55}MnO_3$ while $V_G = +2.5$ V was applied for $SrCoO_{2.5}$, in each case for 15 min, using the same probe station as for samples ST-A and -B. After gating the IL, resist and paraffin were removed using acetone and isopropanol and then CAFM measurements were carried out (see Supplementary Fig. 14).

**Sample characterization.** In situ TEM gating experiments were carried out using a specialized chip with Pt electrodes for applying $V_G$ that was placed within a DENS solution sample holder. An FEI Titan 80-300 microscope, which is probe corrected to achieve a point-to-point resolution of $\sim 1$ Å, was used for the in situ TEM studies. Samples with a [100] in-plane orientation were used. Cross-sectioned TEM

lamellae were prepared from the thin-film samples by conventional grinding and ion-milling techniques. The cross-sectioned lamellae were then transferred onto the in situ chips with electrodes contacting the ends of the lamella as shown in Supplementary Fig. 2. Finally, droplets of the IL were precisely dragged by an atomic force microscope tip to a place close to but not within the region to be imaged (distance $< 5$ µm). The IL was contacted with one of the voltage electrodes ($V_G+$).

Ex situ STEM (HAADF and ABF) were performed in an FEI Titan G2 80-200 ChemiSTEM microscope equipped with an XFEG (Ultra-stable Schottky field emitter gun) and a probe Cs corrector. The microscope was operated at 200 kV. The convergence semi-angle for STEM-HAADF imaging was approximately 22 mrad, while the collection semi-angle was 70–176 or 200 mrad for STEM-HAADF imaging, and 12–24 mrad for STEM-ABF imaging. The thickness of the thin area where we acquired the images is calculated to be 20–25 nm from the zero-loss peak (ZLP) in EELS. All the STEM-HAADF were collected from similar thickness lamellae. EELS and EMCD measurements were performed in a FEI Titan 80-300 STEM equipped with a monochromator unit, a probe spherical aberration corrector, a post-column energy filter system (Gatan Tridiem 865 ER) and a Gatan 2k slow scan CCD system, operating at 300 kV. In the EELS and EMCD measurements, the experiment was performed in the STEM mode, the beam size used was 1.0–2.0 nm and the semi-convergence angle was about 0.5 mrad and the semi-collection angle was about 1.85 mrad. Here samples were [110] in-plane oriented. All the (S)TEM studies were carried out at room temperature, except for EMCD ($\sim 95$ K).

The CAFM function in a Cypher (Asylum Research) atomic force microscopy was used to measure the current flowing across the sample (perpendicular to the surface). A 1 MΩ resistor was connected in series with the $VO_2$ device and a 500 MΩ resistor was connected in series with the $La_{0.45}Sr_{0.55}MnO_3$ and $SrCoO_{2.5}$ devices. An amplifier that allowed for a current measurement resolution of $\sim 1.5$ pA was used at a constant voltage of 1 V. A silicon tip with a Ti/Ir coating (Asyelec-01) was used and the radius of the tip was $\sim 28 \pm 10$ nm. Gating experiments on the Hall bar devices were carried out in a Quantum Design DynaCool at a pressure of a few mTorr of He. The magnetic properties were measured using a Quantum Design superconducting quantum interference device (SQUID). XAS measurements were carried out at Beamline BL08U at the Shanghai Synchrotron Radiation Facility.

**Oxygen level simulations.** A two-dimensional diffusion simulation model was developed based on Fick's second law. In the model anisotropic diffusion coefficients were included. A $O^{2-}$ or $V_O$ injection source with a rate of $R_{inj}$ was introduced to simulate the IL gating effect of negative and positive $V_G$, respectively. The initial BM SCO phase is converted in our model to the perovskite phase when the local normalized O-level exceeds 0.5 (corresponding to $x = 2.75$ for $SrCoO_x$[27]), giving rise to a BM/perovskite boundary in the simulation. We assume that oxygen can diffuse across this boundary only when the O-level difference across the interface exceeds a threshold value $\delta_t$. At the phase boundary there is likely band bending, which will lead to a local excess or deficiency in oxygen concentration[26]. This, we assume, will lead to an asymmetry in oxygen transport across the phase boundary and the diffusion coefficients at the phase boundary are thus modified by $\Delta D$. All the parameters used in the simulations are shown in Supplementary Fig. 18 and Supplementary Table 1.

**Data availability.** The data that support the findings of this study are available from the corresponding author upon reasonable request.

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

## Acknowledgements

The authors thank Lei Jin and Fengshan Zheng from the Jülich Research Center, and Binglun He from Tsinghua University for help with EMCD measurements. The authors acknowledge Beamline BL08U1A in Shanghai Synchrotron Radiation Facility (SSRF) for XAS measurements. We acknowledge partial funding from the EU H2020 program "Phase Change Switch". B.C. thanks the Alexander von Humboldt Foundation for their support. Z.C.W. and X.Y.Z. are thankful for the support of the National Natural Science Foundation of China (51671112, 51471096), National Key Research and Development Program (2016YFB0700402), National Basic Research Program of China (2015CB921700), Tsinghua University (20141081200) and the program "Strategic Partnership RWTH-Aachen University and Tsinghua University".

## Author contributions

B.C. prepared the samples. B.C. and P.W. carried out the in situ TEM measurements. T. M. and B.C. did the simulations. Z.W. carried out the ex situ STEM, EELS, and EMCD under the supervision of X.Z. B.C. carried out all the other measurements. S.S.P.P. conceived and directed the project. All authors participated in discussing the data and writing the manuscript.
