## [Peer Review File · Nature Communications]

Reviewer #1 (Remarks to the Author):

This paper illustrates phase transformation between brownmillerite SrCoO_{2.5} and perovskite SrCoO₃ using the ionic liquid gating. In-situ and Ex-situ TEM techniques are utilized to directly observe crystal structure transformation and to estimate horizontal and vertical transformation velocity, which is highly anisotropic. The possibility of generating 3-D metallic structure using this transformation via IL gating is also discussed, where IL gating is carried out using a patterned resist mask and analyzed using CAFM and corresponding O diffusion simulations.

This is an excellent paper. I really enjoyed reading it. This is the best paper I reviewed in last 6 months. All techniques discussed here are right techniques for these experiments and they are all well executed. I believe this manuscript should be accepted for publication in Nature Communications. A few minor corrections/suggestions are listed below.

While, the manuscript is excellent, to make it better, there are a few points that the authors need to clarify regarding experimental setup and interpretation of the results.

1. The manuscript will benefit from English grammar corrections.
2. Regarding the estimation of the vertical (perpendicular to the surface) transformation velocity, how did the authors obtain ~ 2.9 and ~ 3.6 nm/min? This part needs to be explained in more detail.
3. In Fig. S9, D1/D2 was calculated only from two diffraction spots. Sample misorientation can influence the intensity of spots. What happens if the average intensity of four equivalent diffraction spots is used instead?
4. In the plot, why does the D1/D2 in Fig. S9a, 9b, and 9c decrease at area 9 and 10?
5. From the oxygen diffusion simulation, the transformation velocity could be obtained from the velocity of the phase boundary. How was the result (relative velocity between lateral and horizontal directions) compared to the experimental values?
6. TEM experimental conditions need to be specified. The authors utilize various TEM techniques, including ADF-STEM, ABF-STEM, CTEM, and DP. Especially for the STEM experiment, experimental conditions, i.e. beam energy, convergence angle, detector inner/outer angles, and TEM specimen thickness, are important for reproducibility of results and for determining contributions of z-contrast and strain contrast.
7. EELS experimental conditions need to be clarified. Acquisition area/mode, collection angle, etc.?

Reviewer #2 (Remarks to the Author):

The authors report on a thorough study of structural changes induced by ionic liquid gating in $\text{SrCoO}_{2.5}$ due to controlled removal and incorporation of oxygen vacancies. Despite a complete characterization of the transformation from the oxygen vacancy ordered phase brownmillerite $\text{SrCoO}_{2.5}$ to the oxygen ordered phase perovskite SrCoO_3 in bulk samples, they have been able to perform an original and interesting direct imaging of the reversible transformation by using in-situ high resolution transmission electron microscopy, EELS and EMCD. Furthermore, they use their findings of anisotropic oxygen diffusion induced by ionic liquid gating to design and create 3D mesostructures. They also compare the results on SrCoO_x samples with similar experiments performed in VO_{2-x} and LSMO. The manuscript is well written, the results are novel and the conclusions are sound. The work is very detailed and will be very useful for other researchers in the field. I recommend its publication as is.

We very much appreciate the positive evaluations of our manuscript (NCOMMS-18-03823) by Reviewer #1: “This is an excellent paper. All techniques discussed here are right techniques for these experiments and they are all well executed.” and Reviewer #2: “The manuscript is well written, the results are novel and the conclusions are sound. The work is very detailed and will be very useful for other researchers in the field.”

We address the issues raised by the referees below.

Response to Reviewer #1

Q1. The manuscript will benefit from English grammar corrections.

A: We have read the manuscript carefully and further polished the English with corrections to the grammar.

Q2. Regarding the estimation of the vertical (perpendicular to the surface) transformation velocity, how did the authors obtain ~ 2.9 and ~ 3.6 nm/min? This part needs to be explained in more detail.

A: We have added the following sentence to explain how we estimated the transformation velocities in the main text (Page 5 Line 7 from the bottom): From Fig. 2c-i, the phase boundary moves ~ 20.3 nm over a 7 min period when $V_G = -3$ V, while from Fig. 2k-o, the phase boundary moves ~ 18.0 nm over 5 min when $V_G = +2.5$ V.

Q3. In Fig. S9, D_1/D_2 was calculated only from two diffraction spots. Sample misorientation can influence the intensity of spots. What happens if the average intensity of four equivalent diffraction spots is used instead?

A: We thank the Reviewer for this comment. To avoid the possible influence from misorientation, we chose four equivalent diffraction spots (called D_{BM1-4} , and D_{P1-2}) and calculated the ratio again. The tendency of ratio (with error bar) are shown in Page 13 Line 7 from the bottom of Supplementary Information and new Supplementary Figure 9: The amount of brownmillerite SCO can be qualitatively reflected by the ratio of D_{BM}/D_T . The higher the D_{BM}/D_T ratio, the more brownmillerite $SrCoO_{2.5}$ phase is present. The D_{BM} is the intensity of FFT diffractograms caused by brownmillerite phase alone, while the D_T is the intensity of FFT diffractograms caused by both brownmillerite and perovskite

phases. The depth-dependent phase transitions in Supplementary Figure 9a–e are clearly displayed by the evolution of the ratios, $D_{BM}/D_P = (D_{BM1} + D_{BM2} + D_{BM3} + D_{BM4}) / 2 \times (D_{T1} + D_{T2})$, which are shown in Supplementary Figure 9f–j. The D_{BM1-4} and D_{T1-2} are the intensity of equivalent FFT spots that arise from the brownmillerite phase and both phases, respectively (see the red and blue arrows in Supplementary Figure 9a).

Supplementary Figure 9 | Depth-dependent evolution of SrCoO_x phases.

Depth-dependent line-scan FFTs of TEM images from Figure 2 in the main text: **a**, $V_G = -3$ V, 5 min; **b**, $V_G = -3$ V, 8 min; **c**, $V_G = -3$ V, 10 min; **d**, $V_G = +2.5$ V, 3 min; **e**, $V_G = +2.5$ V, 5 min. **f–j**, Corresponding ratio of D_{BM}/D_T in **a–e**. The error bar of D_{BM}/D_T is the standard deviation for the following four values: D_{BM1}/D_{T1} , D_{BM2}/D_{T1} , D_{BM3}/D_{T2} , and D_{BM4}/D_{T2} . The scale bar is 5 nm.

Q4. In the plot, why does the D1/D2 in Fig. S9a, 9b, and 9c decrease at area 9 and 10?

A: We add the reason for the decreased brownmillerite phase ratio in areas 9 and 10 in Fig. S9a-c in Page 14 Line 6 of Supplementary Information. Although the values of $D_{\text{BM}}/D_{\text{T}}$ for areas 9 in Supplementary Figure 9f-h are slightly lower than those of areas 8, this change in intensity is within the error bar of our measurements. On the other hand, the value of $D_{\text{BM}}/D_{\text{T}}$ is clearly reduced in area 10, but this is because part of the perovskite SrTiO_3 substrate is included in this region of the TEM lamella. Thus, to avoid any misunderstanding, we use open rather than filled symbols for area 10 and use a dashed line to connect these data points with those for area 9 in Supplementary Figure 9f-j.

Q5. From the oxygen diffusion simulation, the transformation velocity could be obtained from the velocity of the phase boundary. How was the result (relative velocity between lateral and horizontal directions) compared to the experimental values?

A: We add the comparison of phase transition anisotropy in experiment and simulation in Page 32 Line 9 of Supplementary Information: In the simulations for SCO, we find that the phase boundary moves ~ 40.4 times faster laterally than vertically, which is comparable to our experimental finding of ~ 34.5 times.

Q6. TEM experimental conditions need to be specified. The authors utilize various TEM techniques, including ADF-STEM, ABF-STEM, CTEM, and DP. Especially for the STEM experiment, experimental conditions, i.e. beam energy, convergence angle, detector inner/outer angles, and TEM specimen thickness, are important for reproducibility of results and for determining contributions of z-contrast and strain contrast.

A: We add the details of STEM experiment conditions in Methods of main text (Page 14 Line 13): *Ex-situ* STEM (HAADF and ABF) were performed in an FEI Titan G2 80-200 ChemiSTEM microscope equipped with an XFEG (Ultra-stable Schottky field emitter gun) and a probe Cs corrector. The microscope was operated at 200 kV. The convergence semi-angle for STEM-HAADF imaging was approximately 22 mrad, while the collection semi-angle was 70-176 or 200 mrad for STEM-HAADF imaging, and 12-24

mrاد for STEM-ABF imaging. The thickness of the thin area where we acquired the images is calculated to be 20–25 nm from the zero-loss peak (ZLP) in EELS. All the STEM-HAADF were collected from similar thickness lamellae.

Q7. EELS experimental conditions need to be clarified. Acquisition area/mode, collection angle, etc.?

A: We add the details of EELS experiment conditions in Methods of main text (Page 14 Line 5 from the bottom): EELS and EMCD measurements were performed in a FEI Titan 80-300 STEM equipped with a monochromator unit, a probe spherical aberration corrector, a post-column energy filter system (Gatan Tridiem 865 ER) and a Gatan 2k slow scan CCD system, operating at 300 kV. In the EELS and EMCD measurements, the experiment was performed in the STEM mode, the beam size used was 1.0–2.0 nm and the semi-convergence angle was about 0.5 mrad and the semi-collection angle was about 1.85 mrad.

Response to Reviewer #2

Thanks to the positive evaluation of “The manuscript is well written, the results are novel and the conclusions are sound. The work is very detailed and will be very useful for other researchers in the field. I recommend its publication as it is. ” and there is no modification in the manuscript due to this evaluation.